# Nutritional Status Evaluation and Intervention in Chronic Kidney Disease Patients: Practical Approach

**DOI:** 10.3390/nu17203264

**Published:** 2025-10-17

**Authors:** Donghyuk Kang, Sojung Youn, Ji Won Min, Eun Jeong Ko

**Affiliations:** Division of Nephrology, Department of Internal Medicine, Bucheon St. Mary’s Hospital, College of Medicine, The Catholic University of Korea, 327, Sosa-ro, Bucheon-si 14647, Gyeonggi-do, Republic of Korea; mc_dong@hanmail.net (D.K.); sjddestar@gmail.com (S.Y.); blueberi12@gmail.com (J.W.M.)

**Keywords:** chronic kidney disease, malnutrition

## Abstract

Malnutrition is a common and serious complication in patients with chronic kidney disease (CKD), significantly impacting morbidity, mortality, and health-related quality of life. Recognizing the close association between nutritional status and clinical outcomes, recent clinical practice guidelines have emphasized proactive nutritional assessment and individualized intervention as key components of CKD management. Accurate nutritional assessment remains challenging in CKD patients due to fluid shifts, altered body composition, and laboratory variability. While various nutritional assessment tools are available, their interpretation requires careful consideration due to each tool’s characteristics and patient’s CKD stage. Nutritional interventions must be tailored to the patient’s CKD stage, dialysis status, and comorbidity profile. Strategies include individualized dietary counseling and oral nutritional supplements. Also, patient education and multidisciplinary collaboration—particularly involving nephrologists and renal dietitians—are essential to improving adherence and long-term outcomes. This review focuses on evaluating the nutritional status and intervention of CKD patients, highlighting key aspects based on the latest literature and clinical guidelines.

## 1. Introduction

Malnutrition is very common in the chronic kidney disease (CKD) population [1]. This is because patients with CKD often experience reduced appetite due to uremia, chronic inflammation, and metabolic abnormalities, and it is difficult for them to maintain a balanced diet due to dietary restriction such as limitations on protein, potassium, phosphorus and sodium intake. Moreover, protein breakdown is increased while protein synthesis is decreased in the CKD population [2,3]. And various metabolic changes such as insulin resistance and heightened inflammatory responses further worsen nutritional status [4]. In addition, dialysis patients may lose protein and water-soluble vitamins during the dialysis process. Furthermore, comorbid conditions such as diabetes and heart failure, along with the medications used to manage them, can also negatively affect nutritional status [5,6,7].

Nutritional status affects the progression of CKD, and conversely, the progression of CKD also impacts nutritional status. Moreover, nutritional status plays a critical role in the clinical outcomes of CKD patients, including mortality and hospitalization rates. Therefore, proper management of nutrition is essential [8].

In this review, we outline a practical overview of nutritional assessment and intervention strategies for patients with CKD, grounded in the latest evidence and clinical guidelines.

## 2. Complex Relationship Between Malnutrition and Clinical Outcomes in CKD Patients

Malnutrition is a prevalent and critical concern in patients with CKD, intricately linked to disease progression and adverse clinical outcomes [9]. As kidney function declines, patients often experience a combination of factors—such as reduced appetite, dietary restrictions, inflammation, metabolic acidosis, and gastrointestinal symptoms—that contribute to inadequate nutritional intake and nutrient loss [10]. This complex interplay results in a high risk of protein-energy wasting (PEW), a condition marked by the loss of body protein and energy reserves, which significantly compromises immune function, physical performance, and overall quality of life [11,12,13].

The consequences of malnutrition in CKD are profound and multifaceted. Numerous studies have shown the malnourished CKD patients, particularly those undergoing dialysis, face increased rates of hospitalization, longer recovery times, higher infection risk, and higher all-cause mortality [14]. Moreover, malnutrition contributes to muscle wasting and frailty, both of which are strong predictors of poor clinical outcomes, including falls, disability, and cardiovascular events. Inflammation, commonly present in CKD, further exacerbates malnutrition by increasing catabolism and impairing appetite, creating a vicious cycle often referred to as the malnutrition-inflammation complex syndrome (MICS) [15,16,17,18].

Addressing malnutrition in CKD requires a comprehensive and multidisciplinary approach, identifying patients at risk and guiding timely interventions. However, effective management must also consider underlying causes such as inflammation, fluid imbalance, and psychosocial factors. Given the strong association between nutritional status and clinical outcomes, early recognition and targeted nutritional strategies are essential components of optimal CKD care.

## 3. Nutritional Assessment Tool

The 2020 Kidney disease Outcomes Quality Initiative (KDOQI) guidelines recommend assessing the nutritional status of CKD patients at least twice year after stage 3, using six key components: anthropometric measurements, laboratory measurements, functional tests such as handgrip strength, measurement of energy requirements, composite nutritional indices, and assessment of protein and calorie intake [19]. However, there is insufficient evidence to recommend one screening tool over another [20]. The commonly used tools for assessing nutritional status in patients with CKD and their characteristics are summarized in Table 1 [20,21,22].

(1)
*To assess body composition: Bioeletrical impedance*


Bioelectrical Impedance Analysis (BIA) is a non-invasive, rapid, and reproducible method widely used to assess body composition—including lean mass, fat mass, and hydration status—in CKD patients. Its clinical value lies in detecting PEW and fluid imbalances, which are critical determinants of morbidity and mortality in this population. The utility and interpretation of BIA vary according to CKD stage and dialysis status. In early CKD stages 1–2, BIA can identify subtle decreases in phase angle (PA)—a raw parameter reflecting cell membrane integrity and hydration—where values below 6° are associated with worse outcomes [23]. As CKD progresses to stages 3–4, fluid overload becomes more prevalent, necessitating cautious interpretation of BIA data; a PA below 5.0° and lean tissue index below the 10th percentile for age and sex indicate malnutrition and muscle wasting [19]. Non-dialysis stage 5 CKD patients pose further challenges due to persistent volume overload and inflammation, making repeated BIA measurements essential to track trends rather than relying on single absolute values. In dialysis patients, BIA is most accurate when performed 30 to 60 min post-dialysis to avoid overestimation of lean mass caused by excess extracellular fluid, with dry weight estimations typically within 1.5 kg of clinical assessments [24].

In patients with limb amputations, standard whole-body BIA protocols are unreliable due to altered body geometry and the loss of conductive tissue pathways, which can underestimate lean mass by 10–15% [25]. Thus, segmental BIA and the application of population-specific correction factors—approximately 5.9% for below-knee and 10.4% for above-knee amputations—are recommended to improve accuracy [19,26]. For amputees undergoing dialysis, BIA measurements should be performed in a euvolemic state, preferably post-dialysis, to minimize the confounding effects of fluid shifts. Serial assessments rather than single time points provide better clinical insight into nutritional status. Due to these complexities, BIA data in amputees should always be interpreted alongside other nutritional assessments such as handgrip strength, Subjective Global Assessment (SGA), and biochemical markers to accurately diagnose protein-energy wasting and guide tailored interventions [24].

Overall, BIA remains a valuable adjunct in the nutritional management of CKD patients, with careful attention to CKD stage, dialysis timing, and patient-specific factors like amputation ensuring optimal clinical application.

(2)
*Subjective Global Assessment*


Subjective Global Assessment (SGA) is a validated clinical tool that integrates patient history and physical examination to classify nutritional status as well-nourished (A), moderately malnourished (B), or severely malnourished (C) by evaluating weight changes, dietary intake, gastrointestinal symptoms, functional capacity, and physical signs of muscle and fat loss [27]. Its simplicity, cost-effectiveness, and bedside applicability make it highly valuable in CKD populations, where malnutrition prevalence ranges from approximately 20% in early stages to over 40% in dialysis-dependent patients [19].

In early CKD stages 1–2, SGA detects subtle nutritional disturbances that may not yet be reflected in biochemical or anthropometric data, such as weight loss exceeding 5% over 3–6 months or decreased food intake below 75% of the usual for more than a week, enabling early nutritional intervention [20]. In stages 3–4, where uremic symptoms like anorexia, nausea, and fatigue become more prevalent, moderate to severe SGA scores correlate with a two- to threefold increase in hospitalization and mortality risk, with weight loss exceeding 10% over six months marking a poor prognosis [28]. For non-dialysis stage 5 patients, fluid retention complicates assessment, necessitating careful interpretation of physical signs such as temporal muscle wasting or subcutaneous fat loss to avoid underestimation of malnutrition.

In dialysis patients, where catabolic processes and inflammation accelerate nutritional decline, SGA is recommended every three months to monitor status. Scores above 6 on the 7-point SGA scale associate with increased mortality risk [22]. Conducting SGAs on non-dialysis or post-dialysis days minimizes confounding from fluid overload. In special populations such as cancer patients, SGA captures symptomatology (e.g., anorexia, early satiety) providing superior prognostic value over body mass index (BMI) alone [29].

Strengths of SGA include its practicality, multidimensional assessment of nutrition, and validation across CKD stages, allowing for repeated, low-cost bedside screening. Limitations stem from its subjective nature, potential interobserver variability, and reduced precision compared to quantitative tools. However, structured training and standardized scoring systems enhance reproducibility and clinical utility, making SGA a cornerstone of nutritional evaluation in CKD when combined with objective measures [30].

(3)
*Malnutrition Inflammation Score*


Malnutrition-Inflammation Score (MIS) is a comprehensive tool developed to assess both nutritional and inflammatory status, particularly in dialysis patients with CKD. Building on the SGA, MIS incorporates additional objective parameters including serum albumin, total iron-binding capacity (TIBC), BMI, comorbidities, and functional status, resulting in a multidimensional evaluation. The total score ranges from 0 to 30, with higher scores indicating more severe malnutrition and inflammation [28].

MIS is a strong predictor of adverse clinical outcomes such as hospitalization and mortality, especially in dialysis patients where inflammation is prevalent in 30–50% of cases and exacerbates protein-energy wasting [19]. Scores above 7–8 correlate with a two- to threefold increased risk of mortality and morbidity within 12 months [28]. This prognostic power surpasses that of SGA or serum albumin alone, making MIS particularly valuable for guiding nutritional and anti-inflammatory interventions in this high-risk group.

In non-dialysis CKD patients, especially in advanced stages 4 and 5, MIS aids in differentiating malnutrition driven by inflammation from other causes, although its routine use is limited by the need for frequent laboratory data [19]. An MIS above 6 in these patients signals the need for close nutritional monitoring and intervention.

MIS correlates well with objective markers such as handgrip strength and lean body mass, allowing for cross-validation of nutritional status [31]. However, its complexity, time requirements, and dependence on laboratory tests make it less feasible in resource-limited settings. Furthermore, despite structured training, some subjectivity and inter-rater variability persist [20,28].

Despite its advantages, the MIS is more time-consuming and complex to administer compared to simpler tools such as the SGA. It requires laboratory values and trained personnel for accurate evaluation, which limits its practicality and widespread use outside specialized dialysis centers. Additionally, some components of the MIS still rely on clinical judgment, introducing potential inter-rater variability and subjectivity in the assessment.

Regular reassessment every three months is recommended to monitor disease progression and treatment response [20].

(4)
*Body Mass Index*


Body Mass Index (BMI) is a simple, widely used, and non-invasive tool for assessing nutritional status in patients with CKD. It estimates body fat based on weight and height, serving as an initial screening parameter for undernutrition or obesity [32,33]. In CKD populations, particularly those on dialysis, a low BMI has been linked to increased morbidity and mortality, making it a valuable prognostic indicator [34].

However, BMI has notable limitations in this context. It does not differentiate between fat mass and lean body mass, nor does it account for fluid overload, which is common in CKD and can mask true body composition [35]. Consequently, BMI may underestimate or overestimate nutritional status in patients with edema or muscle wasting [36]. Furthermore, BMI does not capture dynamic nutritional changes over time and lacks information on micronutrient status or inflammation. Despite these limitations, BMI remains a convenient and accessible initial screening tool when combined with clinical, biochemical, and functional assessments to provide a more comprehensive evaluation of nutritional status in CKD patients.

In dialysis patients, fluid shifts and overload can confound BMI interpretation, often causing overestimation of adiposity [19]. BMI should ideally be measured post-dialysis or at dry weight to improve accuracy. Notably, dialysis patients exhibit an “obesity paradox,” where higher BMI (>30 kg/m^2^) associates with better survival compared to underweight individuals (BMI < 18.5 kg/m^2^), who face a twofold increased mortality risk [37]. Because BMI alone cannot distinguish muscle from fat mass, complementary methods like BIA or handgrip strength (HGS) are recommended.

For non-dialysis CKD patients, BMI interpretation is more reliable due to less pronounced fluid overload but still requires caution, especially in advanced stages 4–5 [20]. A BMI below 23 kg/m^2^ is associated with higher risk of progression to end-stage kidney disease and mortality, particularly when combined with unintentional weight loss exceeding 5% over six months. Conversely, obesity (BMI ≥ 30 kg/m^2^) contributes to CKD progression through metabolic complications such as insulin resistance and inflammation. Therefore, BMI cutoffs in CKD populations may differ from general guidelines and should be interpreted alongside other nutritional markers and clinical context.

While limited by its inability to distinguish between fat, muscle, and fluid, BMI remains a practical, inexpensive, and widely available tool suitable for longitudinal monitoring. Regular BMI assessment every 1–3 months helps detect early nutritional deterioration or excessive weight gain, enabling timely interventions when used in conjunction with functional and biochemical evaluations.

(5)
*Handgrip strength*


Handgrip Strength (HGS) is a simple, non-invasive, and cost-effective tool for assessing muscle strength and nutritional status in patients with CKD [38,39]. As a direct measure of muscle function, HGS is particularly valuable in identifying sarcopenia and malnutrition, conditions commonly seen in CKD [40]. It correlates strongly with clinical outcomes including hospitalization, morbidity, and mortality, and can detect early muscle function decline not apparent through anthropometric or biochemical measures [17].

In dialysis patients, HGS is typically measured on the non-fistula arm before dialysis to avoid the effects of fatigue and fluid shifts [19]. Cutoff values below 27 kg for men and 16 kg for women are widely accepted to diagnose clinically significant muscle weakness and sarcopenia [24]. Patients below these thresholds have a 1.5- to 3-fold increased risk of hospitalization and mortality within 12 months, independent of BMI or serum albumin. HGS also responds to nutritional and exercise interventions, with improvements of 2–3 kg commonly observed after targeted therapies [19].

In non-dialysis CKD patients, low HGS predicts faster progression to end-stage kidney disease (ESRD) and increased mortality, often preceding detectable changes in body composition. Additionally, HGS is useful in monitoring frailty, especially in elderly CKD populations where frailty prevalence can reach 50%, aiding in personalized rehabilitation [24].

Limitations of HGS include its dependency on patient effort, influence by hand dominance, and comorbidities such as arthritis or neuropathy, which may affect reliability. Standardized protocols recommend taking the highest of three attempts and adjusting for age and sex norms to improve reproducibility [39]. Despite these limitations, HGS remains a vital, objective biomarker that complements other nutritional and functional assessments in CKD.

(6)
*Serum albumin/pre-albumin*


Serum albumin and pre-albumin are commonly used biochemical markers in assessing nutritional status in patients with CKD, due to their association with protein-energy wasting and clinical outcomes [31]. Serum albumin, with a longer half-life (~20 days), reflects chronic nutritional status and inflammation, consistently predicting morbidity and mortality in both dialysis and non-dialysis CKD populations [17]. Levels below 3.8 g/dL in dialysis patients correlate with increased hospitalization and mortality risk, with each 0.1 g/dL decrease linked to a 10–15% higher risk of death [28].

Pre-albumin (transthyretin), with a shorter half-life (~2 days), is more sensitive to acute nutritional changes and can be used to monitor the effectiveness of nutritional interventions over shorter periods. Values below 30 mg/dL in dialysis patients are associated with malnutrition and worse outcomes. Pre-albumin is less affected by hydration status but remains sensitive to inflammation, limiting its specificity [21,22].

Both markers, however, have important limitations. They are influenced by non-nutritional factors such as inflammation, hydration status, liver function, and infections, which are prevalent in CKD and complicate interpretation. For example, hypoalbuminemia often reflects systemic inflammation rather than pure malnutrition. Their levels may also remain normal despite significant muscle wasting or inadequate dietary intake. Therefore, serum albumin and pre-albumin should be interpreted in conjunction with clinical evaluation, dietary history, inflammatory markers like C-reactive protein (CRP), and other nutritional assessments to ensure accurate diagnosis [31].

In non-dialysis CKD, hypoalbuminemia (<3.5 g/dL) predicts faster progression to end-stage kidney disease and increased cardiovascular mortality, independent of eGFR. Serial measurements help detect clinically significant declines (>0.2 g/dL over 3 months) indicating worsening nutritional or inflammatory status.

Special populations, such as frail elderly or cancer-associated CKD patients, often present with hypoalbuminemia driven by combined inflammation, protein catabolism, and reduced intake, correlating with poor prognosis. Despite their limitations, serum albumin and pre-albumin remain valuable prognostic markers and components of multifactorial nutritional assessment in CKD, with recommended monitoring intervals every 1–3 months [20].

(7)
*Normalized protein catabolic rate*


Normalized protein catabolic rate (nPCR) is a biochemical tool primarily used in dialysis patients to estimate daily protein intake by calculating urea nitrogen generation normalized to body weight (g/kg/day), serving as an indirect marker of dietary protein consumption and nutritional status [19]. A target nPCR ≥ 1.0 g/kg/day is recommended in dialysis patients to maintain nitrogen balance, while values below 0.8 g/kg/day indicate insufficient protein intake and increased risk of PEW, correlating with higher morbidity and mortality.

nPCR is derived from pre- and post-dialysis blood urea nitrogen (BUN) measurements, dialysis adequacy (Kt/V), and patient weight, making it highly useful for monitoring and guiding individualized nutritional interventions over time. In peritoneal dialysis, adjustments for continuous dialysate clearance and residual renal function are necessary for accurate calculation.

However, nPCR assumes steady-state metabolic and dialysis conditions, which may not apply in acutely ill or unstable patients, potentially limiting reliability. Additionally, nPCR can be confounded by non-nutritional factors such as inflammation, catabolic stress, and residual renal function, possibly leading to over- or underestimation of true protein intake. Therefore, it should be interpreted alongside inflammatory markers such as C-reactive protein (CRP) to distinguish malnutrition from hypercatabolic states [21].

In non-dialysis CKD patients, nPCR is less commonly used due to the absence of dialysis-related measurements, but 24 h urine urea nitrogen can estimate protein intake, with targets around 0.8–1.0 g/kg/day depending on CKD stage and comorbidities. Low nPCR (<0.8 g/kg/day) in advanced CKD stages signals inadequate protein intake and warrants nutritional support [20].

Limitations of nPCR include dependence on accurate BUN and dialysis adequacy data, inability to assess energy intake or micronutrient status, and potential inaccuracies during non-steady-state conditions. Despite these constraints, nPCR remains a valuable, objective marker for assessing protein intake and guiding nutritional management in dialysis-dependent CKD patients, with recommended monitoring every 1–3 months [19].

(8)
*GLIM criteria*


The Global Leadership Initiative on Malnutrition (GLIM) criteria diagnose malnutrition through combined phenotypic (weight loss, low BMI, reduced muscle mass) and etiologic factors (reduced food intake/assimilation and inflammation), requiring at least one criterion from each category. This framework effectively addresses the multifactorial nature of malnutrition in CKD [41]

In dialysis patients, malnutrition prevalence by GLIM criteria is approximately 30–50%, strongly associated with increased hospitalization and mortality (Phenotypic thresholds include unintentional weight loss > 5% over 6 months, BMI < 20 kg/m^2^ (<70 years) or <22 kg/m^2^ (≥70 years), and reduced muscle mass assessed via bioelectrical impedance analysis (BIA) or dual-energy X-ray absorptiometry (DXA)). Etiologic factors emphasize reduced food intake (<50% of energy needs for >1 week) and inflammation, which affects up to half of dialysis patients [19].

In non-dialysis CKD, GLIM is less commonly used but remains applicable; weight loss and reduced intake serve as primary indicators, with inflammation gaining importance in advanced stages 3–5. Nutritional interventions guided by GLIM have been shown to improve outcomes and slow protein-energy wasting [20].

CKD stage-specific considerations include:

*Stages 1–2*: Detection mainly through weight loss > 5% in 6 months and reduced intake (<50% energy needs for >1 week), with inflammation playing a lesser role.

*Stages 3–4*: Inflammation becomes a more prominent etiologic factor; muscle mass assessment using BIA or handgrip strength improves early detection of malnutrition.

*Stage 5 (non-dialysis and dialysis)*: Frequent inflammation and fluid overload make GLIM particularly valuable, with phenotypic criteria including BMI cutoffs, weight loss, and muscle mass measurement via BIA or DXA. Assessment every 3–6 months is recommended.

Limitations of GLIM include the need for validated muscle mass measurement tools and the potential subjectivity in estimating reduced food intake, which may restrict its implementation in some clinical settings.

## 4. Nutritional Intervention

(1)
*Energy (Calorie) Intake*


Maintaining adequate energy intake is a cornerstone in the nutritional management of CKD to prevent PEW, a syndrome characterized by loss of body protein and fat stores that significantly increases morbidity, hospitalization, and mortality across all CKD stages [42]. Kidney Disease Improving Global Outcomes (KDIGO) 2024 guidelines emphasize tailoring energy intake to meet individual metabolic demands, thereby preventing catabolism and ensuring that dietary protein supports anabolic functions such as tissue repair and immune response rather than serving as an energy substrate [20].

Energy requirements vary across CKD stages and according to dialysis status, reflecting differences in metabolic rate, inflammation, and fluid balance. In early CKD stages 1–3, KDIGO, KDOQI, European society of clinical nutrition and metabolism (ESPEN), and International Society of Renal Nutrition and Metabolism (ISRNM) uniformly recommend energy intake of 30–35 kcal/kg ideal body weight per day, given relatively stable metabolic conditions and minimal catabolic stress. However, adjustments should be made based on age, physical activity, and comorbidities; older adults and sedentary patients may require intake at the lower end of this range to avoid overnutrition.

In stages 4–5 CKD without dialysis, energy needs are slightly more variable due to increasing metabolic derangements and often chronic fluid overload. Guidelines suggest maintaining 25–35 kcal/kg/day, with careful monitoring to balance the risks of undernutrition and overnutrition (Table 2).

In dialysis populations, metabolic demands and protein turnover increase due to dialysis-related inflammation, catabolic stimuli, and loss of amino acids during treatment. Consequently, higher energy intake is necessary. KDIGO and KDOQI recommend 30–40 kcal/kg ideal body weight per day in dialysis patients to compensate for increased energy expenditure and to preserve lean body mass and functional status. ESPEN echoes these recommendations, emphasizing the need for individualized assessment given the heterogeneity of dialysis patients, while ISRNM stresses the importance of frequent reassessments to account for fluctuations in clinical status.

Energy provision should prioritize nutrient-dense sources with emphasis on healthy fats (mono- and polyunsaturated fatty acids), complex carbohydrates, and fiber, while minimizing simple sugars and ultra-processed foods to reduce metabolic stress. For patients unable to meet energy needs through diet alone, oral nutritional supplements (ONS), preferably high in protein and energy, are recommended to prevent or reverse weight loss and frailty. Increasing meal frequency to five to six small meals or snacks per day, rather than the traditional three large meals, can improve overall intake without causing gastrointestinal discomfort. Energy-dense foods such as nut butters, avocados, full-fat dairy products, and healthy oils (olive or canola) are effective strategies to enhance caloric intake efficiently.

Addressing modifiable factors that impair appetite is critical to optimize energy intake. Correction of metabolic acidosis with oral bicarbonate improves appetite and muscle preservation, while management of chronic inflammation, uremic toxins, and depression supports patient engagement and adherence to nutritional plans. Psychological and pharmacologic interventions for depression further enhance appetite and dietary compliance.

Early identification of nutritional risk allows for timely interventions. Renal dietitians should provide individualized meal planning that prioritizes energy-dense, nutrient-rich foods to meet caloric goals while maintaining protein restriction. Use of oral nutritional supplements tailored for renal patients can help achieve energy targets without excessive protein or electrolytes Additionally, counseling should emphasize education on appropriate portion sizes, nutrient timing, and strategies to overcome anorexia, such as small frequent meals and flavor enhancement. Multidisciplinary coordination between nephrologists, dietitians, nurses, and pharmacists ensures comprehensive management of comorbidities that may impact nutrition, such as diabetes or cardiovascular disease.

(2)
*Protein Intake*


In patients with CKD, low protein diets (LPDs) are frequently prescribed as a therapeutic strategy to reduce nitrogenous waste products, control uremic symptoms, and slow the progression of renal function decline. This dietary intervention has shown clinical benefits particularly in the pre-dialysis stages; however, it carries a significant risk of malnutrition when not individualized or carefully monitored [21]. Protein is an essential macronutrient necessary for the maintenance of muscle mass, tissue repair, immune function, and the synthesis of vital enzymes and hormones. Given that CKD patients often experience anorexia, systemic inflammation, and increased protein catabolism, further restriction of protein intake may precipitate PEW, a syndrome characterized by the progressive loss of both protein and energy stores [22]. PEW is associated with sarcopenia, compromised immune defenses, increased susceptibility to infections, diminished physical function, and overall poorer clinical outcomes, including higher rates of hospitalization and mortality [23,24,25]. These risks are further amplified in elderly patients and those with comorbidities such as diabetes or heart failure, where nutritional requirements are more complex [26]. Additionally, without appropriate dietary counseling, patients may misinterpret LPDs as a mandate to reduce overall food intake, resulting not only in protein deficiency but also inadequate consumption of calories and micronutrients [27]. Therefore, although LPDs can be beneficial when properly prescribed, they must be implemented with individualized nutritional assessments, regular monitoring, and adequate supplementation—including the potential use of keto acid analogs—to prevent unintended and harmful nutritional deficiencies [28]. A multidisciplinary approach involving nephrologists, dietitians, and patients is critical to balance renal protection with nutritional adequacy effectively.

To prevent malnutrition associated with LPDs in CKD patients, a carefully balanced and individualized nutritional approach is essential. The incorporation of high-quality protein sources that are rich in essential amino acids but low in phosphorus and potassium can optimize nutrient intake and minimize complications related to mineral overload [29]. Effective prevention of PEW necessitates maintaining protein intake targets tailored to the patient’s CKD stage or dialysis modality, while ensuring sufficient daily energy intake, typically 30–35 kcal/kg of body weight. For non-dialysis CKD patients, protein intake is generally recommended at 0.6–0.8 g/kg/day, whereas dialysis patients require higher protein intakes of up to 1.0–1.2 g/kg/day to prevent PEW [19,20]. Supplementation with keto-analogs of essential amino acids has been demonstrated to support protein metabolism without increasing nitrogenous waste production, thus mitigating malnutrition risk while preserving renal function [30]. Clinical studies recommend a daily dosage of keto-analog supplements of approximately 0.1 g/kg of body weight, equivalent to one 500 mg tablet per 5 kg of body weight, administered in divided doses with meals [31,32]. For example, a 70 kg patient would require around 7 g per day or 14 tablets daily. This supplementation maintains adequate amino acid availability, corrects metabolic imbalances, and may delay the initiation of dialysis without compromising nutritional status. Evidence supports keto-analog therapy as a safe and effective strategy to prevent PEW in pre-dialysis CKD stages, provided it is appropriately dosed and carefully monitored [33,34] (Table 2).

Close monitoring by a renal dietitian is crucial to ensure that energy intake remains adequate to spare protein for tissue maintenance and repair. Regular nutritional assessments, including anthropometric measurements, biochemical markers, and clinical status evaluations, enable timely adjustments to diet plans, thus preventing progression of malnutrition [35]. Furthermore, patient education on the importance of dietary compliance and appropriate food choices, combined with multidisciplinary collaboration, is vital for successful nutritional management while maintaining the benefits of a low protein diet.

The 2024 KDIGO clinical practice guideline emphasizes balancing the risks associated with excessive protein load on the kidneys against the dangers of PEW from inadequate intake. For adults with CKD stages 3 to 5 who are not on dialysis, KDIGO recommends a protein intake of approximately 0.8 g/kg body weight per day. This level is sufficient to maintain muscle mass while reducing glomerular hyperfiltration and nitrogenous waste generation, both of which contribute to CKD progression. Diets high in protein (>1.3 g/kg/day) are discouraged for patients at risk of kidney function decline. In selected, motivated patients with stable metabolic status, KDIGO supports considering very low-protein diets (VLPD) of 0.3–0.4 g/kg/day, provided they are supplemented with keto acid analogs or essential amino acids. This intervention may delay the need for dialysis and improve metabolic outcomes but must only be implemented under close clinical and dietetic supervision. Such restrictive approaches are contraindicated in metabolically unstable individuals, including those with active inflammation, catabolism, or significant comorbidities. Moreover, in elderly individuals or those with frailty or sarcopenia, higher protein intake may be necessary to prevent muscle wasting, even if CKD is advanced. These nuanced recommendations reflect KDIGO’s commitment to precision nutrition by adapting protein intake not only to kidney function but also to age, metabolic state, and physiological needs. This approach is further supported by recent trials and expert consensus in nephrology nutrition.

In clinical practice, the implementation of these recommendations requires a structured, stepwise approach. First, early identification of patients at risk of malnutrition through routine nutritional screening—including weight trends, body composition assessment, dietary recall, and functional measures such as handgrip strength—is essential [7,35]. Second, individualized dietary plans should be developed by renal dietitians in collaboration with the healthcare team, tailoring protein and energy targets to the patient’s CKD stage, dialysis status, comorbid conditions, and lifestyle factors [19,20,28]. Third, patients on low or very low protein diets should be considered for keto-analog supplementation, with dosages carefully calculated based on body weight and divided across meals to optimize metabolic effects and adherence [30,31,32,33,34]. Fourth, ongoing nutritional monitoring through periodic biochemical markers such as serum albumin, prealbumin, nPCR, and inflammatory markers alongside clinical assessments is vital to adjust dietary prescriptions in response to changing metabolic or clinical status [6,35]. Finally, multidisciplinary education and psychosocial support must be provided to address barriers to adherence, including appetite changes, food access issues, and patient understanding of dietary goals [10,22]. This comprehensive approach enhances the likelihood of achieving a balance between renal protection and nutritional adequacy, thereby improving clinical outcomes in CKD patients [7,21].

(3)
*Sodium Intake*


Dietary sodium restriction is a critical component of managing CKD due to its well-established effects on blood pressure control, extracellular fluid volume regulation, proteinuria reduction, and ultimately the slowing of CKD progression. Excessive sodium intake contributes to salt-sensitive hypertension and glomerular hyperfiltration, mechanisms that accelerate nephron loss and increase cardiovascular risk in this population.

The 2024 KDIGO Clinical Practice Guideline recommends limiting sodium intake to less than 2 g per day, equivalent to approximately 5 g of salt, for patients across CKD stages 3 to 5, including both non-dialysis and dialysis populations. This recommendation aligns with global cardiovascular health targets and is designed to mitigate hypertension and volume overload while allowing individualized adjustments based on clinical volume status, serum sodium concentrations, and patient symptoms, particularly in elderly or volume-sensitive patients. Similarly, the KDOQI 2020 guidelines advocate for sodium restriction below 2.3 g per day for CKD stages 3 to 5, with additional caution in dialysis patients due to altered fluid balance and the risk of intradialytic hypotension. Peritoneal dialysis patients may require more flexible sodium targets to maintain adequate nutrition and volume control [1]. The ESPEN 2020 guidelines suggest a sodium intake range of 1.5 to 2 g per day in pre-dialysis CKD patients, emphasizing the importance of avoiding both excessive sodium consumption and overly aggressive restriction, especially among elderly or frail individuals who are at higher risk of hyponatremia [2].

Despite these clear numerical targets, achieving effective sodium restriction in clinical practice extends beyond advising patients to “avoid the salt shaker.” Given that 70–80% of sodium intake in industrialized nations derives from processed foods, restaurant meals, and packaged snacks, patient education must focus on identifying hidden sodium sources within common dietary items. Patients should be trained to interpret nutrition labels carefully, recognizing terms such as “sodium,” “salt,” “sodium chloride,” and “monosodium glutamate.” Avoidance of high-sodium processed foods—including canned soups, deli meats, frozen entrees, soy sauce, and certain condiments—is essential to maintaining sodium intake within recommended limits without compromising overall nutrition.

A gradual, staged reduction in sodium intake has been shown to improve patient adherence and minimize adverse effects such as reduced quality of life or hypotension. A practical approach involves decreasing added salt by approximately 25% every two weeks, allowing for taste adaptation. Flavor can be preserved through the use of non-sodium seasonings such as herbs (basil, oregano, thyme), spices (black pepper, paprika), and acidic flavorings (lemon juice, vinegar). Collaboration with renal dietitians is pivotal to developing culturally and regionally appropriate low-sodium meal plans and recipes that emphasize home cooking with fresh, unprocessed ingredients rather than reliance on processed or restaurant foods.

Regular monitoring of blood pressure, volume status (including assessment of edema and weight changes), and serum sodium levels is essential to guide individualized sodium targets and prevent complications such as hyponatremia or intradialytic hypotension, especially in elderly or frail patients. A multidisciplinary team approach involving nephrologists, dietitians, nursing staff, and social workers ensures that sodium restriction strategies are personalized to the patient’s CKD stage, dialysis modality, comorbidities, lifestyle, and preferences, thereby promoting sustainable long-term adherence.

(4)
*Potassium Intake*


The management of dietary potassium in CKD has evolved from universal potassium restriction toward a more nuanced, individualized approach that balances potassium control with overall diet quality and nutritional status. The 2024 KDIGO guideline recommends against routine potassium restriction in CKD patients without persistent hyperkalemia, emphasizing correction of modifiable non-dietary contributors such as metabolic acidosis, constipation, and medications including renin–angiotensin–aldosterone system (RAAS) inhibitors or potassium-sparing diuretics before dietary changes. KDIGO advises maintaining potassium intake consistent with a healthy diet rich in fruits, vegetables, legumes, and whole grains unless hyperkalemia persists, as these foods provide fiber, antioxidants, and alkalinizing effects beneficial for cardiovascular and kidney health.

KDIGO does not prescribe a fixed potassium intake limit but recommends individualized targets based on serum potassium levels and CKD stage. For example, in CKD stages 3–5 without hyperkalemia, potassium intake is generally unrestricted (often 2000–4000 mg/day, consistent with general population guidelines), whereas in advanced stages or dialysis patients with hyperkalemia, selective restriction of high-potassium processed foods and additives is advised [20].

The KDOQI Nutrition Guidelines (2020) provide more specific numeric recommendations, suggesting potassium intake of approximately 2000–3000 mg/day in non-dialysis CKD patients with stable potassium levels. For dialysis patients, intake may be adjusted between 2000 and 3500 mg/day depending on serum potassium, dialysis modality, and residual renal function, balancing risk of hyperkalemia against nutritional adequacy. KDOQI stresses frequent serum potassium monitoring and individualized dietary counseling [19].

ESPEN guidelines similarly recommend maintaining potassium intake within 2000–3000 mg/day for CKD patients without hyperkalemia, allowing liberalized intake of potassium-rich plant foods to promote nutritional quality. In cases of persistent hyperkalemia, a tailored reduction focusing on limiting potassium additives and processed foods is recommended [24].

Clinically, the first step is regular monitoring of serum potassium to guide dietary recommendations. When dietary potassium restriction is warranted, practical strategies include patient education on food preparation techniques that reduce potassium content by up to 50%, such as peeling, cutting into small pieces, soaking for at least 2 h, and boiling in a large volume of water with discard of the cooking liquid. This approach enables patients to continue consuming nutrient-rich vegetables and legumes while managing hyperkalemia risk.

Patients should be counseled to choose lower-potassium fruits—such as apples, berries, and grapes—and limit higher-potassium fruits like bananas, oranges, and avocados when potassium levels are elevated. Developing culturally sensitive and individualized meal plans with the support of renal dietitians improves adherence and nutritional status.

Patient education on timing (typically with meals) and monitoring for adverse effects is essential, with laboratory checks every 2 to 4 weeks following therapy initiation or adjustment.

(5)
*Phosphorus Intake*


Phosphorus control is essential in CKD management due to its pivotal role in CKD-mineral and bone disorder (CKD-MBD), vascular calcification, and cardiovascular risk. The 2024 KDIGO guidelines recommend an individualized approach to dietary phosphorus restriction, guided by CKD stage, serum phosphate levels, and risk of CKD-MBD, rather than universal limitation. Phosphorus bioavailability varies substantially by source: plant-based phosphorus, mostly bound to phytates, has low absorption (<50%), while animal-derived phosphorus has moderate bioavailability (40–60%) In contrast, inorganic phosphate additives found in processed foods exhibit near-complete absorption (90–100%) and substantially elevate serum phosphate, making them key targets for dietary intervention [43].

In early CKD stages 1–3 with normal phosphate levels, strict phosphorus restriction is generally unnecessary. However, in advanced CKD stages 4–5 and dialysis patients, phosphorus intake should be limited to approximately 800–1000 mg/day to mitigate hyperphosphatemia and secondary hyperparathyroidism. KDOQI (2020) aligns with these recommendations, endorsing 800–1000 mg/day for non-dialysis CKD patients with hyperphosphatemia, with adjustments for dialysis populations. ESPEN and ISRNM guidelines similarly recommend phosphorus intakes below 1000 mg/day in advanced CKD and dialysis, individualized by residual kidney function and phosphate binder use.

Clinical management should prioritize patient education to identify and avoid phosphate additives frequently labeled as “phosphate,” “phosphoric acid,” or “polyphosphate” in processed meats, sodas, and baked goods. Practical cooking methods such as overnight soaking and boiling with discarded soaking water can reduce phosphorus in plant foods by 30–50%, enabling continued intake of nutrient-rich legumes and whole grains. Balancing phosphorus restriction with adequate protein intake is critical to prevent protein-energy wasting; thus, monthly monitoring of serum phosphate and nutritional status is essential. Phosphate binders remain a cornerstone for controlling hyperphosphatemia when diet alone is insufficient, with emphasis on taking binders with meals and monitoring adherence and side effects.

When integrating CKD-MBD management, it is vital to consider not only serum phosphate but also calcium levels, parathyroid hormone (PTH), and fibroblast growth factor 23 (FGF23), as these biomarkers inform phosphorus targets and treatment intensity [44]. Elevated phosphate and PTH levels accelerate vascular calcification and bone disease; thus, clinicians must tailor phosphorus intake and binder therapy accordingly to minimize these risks. Moreover, dietary phosphorus restriction must be carefully balanced with calcium intake and vitamin D status to optimize bone health and avoid exacerbating secondary hyperparathyroidism. Recent evidence underscores the importance of avoiding excessive calcium-based phosphate binders in patients with vascular calcification, favoring non-calcium binders where appropriate [45].

A multidisciplinary team approach involving nephrologists, dietitians, pharmacists, and social workers enhances patient adherence and outcomes.

## 5. Special Considerations in Assessing Nutritional Status of CKD Patients

(1)
*Cognitive Assessment in Nutritional Status Evaluation for CKD Patients*


Cognitive impairment is increasingly recognized as a common and clinically significant comorbidity in patients with CKD, particularly in advanced stages and among dialysis patients. The interplay between cognitive decline and malnutrition creates a complex clinical scenario, as impaired cognition can lead to poor dietary adherence, reduced self-care ability, and subsequently worsen nutritional status [46].

Cognitive assessment should be integrated into the comprehensive evaluation of nutritional status in CKD patients to identify those at risk for malnutrition due to decreased ability to manage dietary restrictions and nutrition plans. Standardized screening tools such as the Montreal Cognitive Assessment (MoCA) and Mini-Mental State Examination (MMSE) are commonly employed to detect mild cognitive impairment and dementia in this population. Early identification allows for tailored interventions, including caregiver involvement and simplified nutrition regimens [47].

Several pathophysiological mechanisms contribute to cognitive impairment in CKD, including uremic toxin accumulation, chronic inflammation, anemia, and vascular disease, all of which may also exacerbate malnutrition and frailty. Therefore, cognitive decline and malnutrition share overlapping pathways and should be addressed simultaneously [48].

Practical approaches to integrate cognitive assessment into nutritional management involve multidisciplinary collaboration among nephrologists, dietitians, neuropsychologists, and social workers. For patients with documented cognitive deficits, interventions might include simplified dietary instructions, oral nutritional supplements with easy administration, and enhanced caregiver education to ensure adherence [49].

Recent guidelines emphasize that cognitive screening is an essential component of comprehensive CKD care, as addressing cognitive impairment may improve nutritional outcomes and overall quality of life. Future research is warranted to establish standardized protocols for cognitive and nutritional assessment and to develop targeted interventions that address this dual burden.

(2)
*Nutritional Assessment and Intervention in Frail CKD Patients*


Frailty significantly complicates nutritional assessment and management in CKD patients due to its association with decreased muscle mass, functional impairment, and chronic inflammation [50]. Nutritional evaluation in frail CKD patients should include objective measures of muscle function and quantity, such as HGS with cutoffs below 27 kg for men and 16 kg for women indicating reduced muscle strength, as recommended by the European Society for Clinical Nutrition and Metabolism. Muscle mass assessment via BIA or DXA is essential, with appendicular skeletal muscle mass index (ASMI) thresholds of <7.0 kg/m^2^ for men and <5.5 kg/m^2^ for women signaling sarcopenia, which frequently coexists with frailty. These functional and morphological assessments are crucial, as traditional markers such as BMI or serum albumin are often misleading due to fluid overload and systemic inflammation prevalent in this population [39,51].

Due to frequent anorexia, early satiety, and gastrointestinal symptoms in frail CKD patients, multidisciplinary interventions including appetite stimulants, individualized meal plans enriched with leucine or branched-chain amino acids, and resistance exercise programs are advised to improve protein-energy status and muscle function. Monitoring should be performed every 1–3 months with repeat assessments of muscle strength, dietary intake, and biochemical markers to promptly identify nutritional decline and adjust treatment. This comprehensive and proactive strategy is essential to reduce hospitalization rates, morbidity, and mortality associated with frailty in CKD [19,20,51] (Table 2).

Nutritional intervention in frail CKD patients requires a tailored approach that balances the increased protein needs necessary to combat muscle wasting with the risk of accelerating kidney disease progression. Energy and protein intake should be monitored. Furthermore, chronic inflammation—a hallmark of frailty—can elevate protein catabolism and reduce appetite, requiring regular monitoring of inflammatory markers such as C-reactive protein (CRP), with levels > 5 mg/L indicating active inflammation that may necessitate anti-inflammatory and nutritional support interventions [10,19,20].

(3)
*Nutritional Assessment and Intervention in Malignancy CKD Patients*


Energy requirements in CKD patients with malignancy are generally increased due to the hypermetabolic state associated with cancer. ESPEN guidelines recommend targeting an energy intake of 30–35 kcal/kg/day in stable CKD, increasing to 35 kcal/kg/day or more during active cancer treatment or episodes of inflammation [21]. Protein intake must be carefully balanced: while standard CKD management often limits protein to 0.6–0.8 g/kg/day in non-dialysis stages to reduce kidney workload, patients with malignancy-related PEW require a higher protein provision of at least 1.2 g/kg/day to support muscle maintenance and immune function. In dialysis-dependent patients, a target protein intake of ≥1.2 g/kg/day is advised, with close monitoring to avoid exacerbating azotemia [19,21,52] (Table 2)

Nutritional interventions should incorporate early and aggressive strategies, including oral nutritional supplements enriched with omega-3 fatty acids and branched-chain amino acids, which have shown efficacy in attenuating muscle wasting and inflammation in cancer-CKD populations [53]. In patients unable to meet energy and protein needs orally, enteral nutrition is preferred over parenteral routes to maintain gut integrity and reduce infection risk. Furthermore, pharmacologic adjuncts such as appetite stimulants or anti-inflammatory agents may be considered, although these require individualized risk-benefit analysis given the altered pharmacokinetics in CKD and cancer [21].

Regular nutritional reassessment every 4–6 weeks is crucial in this group due to the dynamic nature of malignancy-associated metabolic changes and the increased risk of rapid deterioration [19]. Multidisciplinary collaboration involving nephrologists, oncologists, dietitians, and palliative care specialists is essential to optimize nutritional status, improve quality of life, and potentially enhance clinical outcomes in CKD patients with malignancy.

(4)
*Nutritional Assessment and Intervention in Multi-tablet therapy*


CKD patients undergoing multi-tablet therapy present unique challenges in nutritional assessment and intervention due to the high risk of drug–nutrient interactions, altered appetite, and impaired nutrient. Polypharmacy, often defined as the concurrent use of five or more medications, is prevalent in CKD populations, with some patients taking upwards of 10–15 tablets daily to manage comorbidities such as hypertension, diabetes, and mineral bone disorders. This complexity necessitates meticulous nutritional evaluation and individualized intervention plans.

Nutritional assessment in this context must incorporate a thorough medication review to identify agents that may adversely affect nutritional status. For example, phosphate binders can interfere with the absorption of fat-soluble vitamins and trace elements, while proton pump inhibitors (PPIs) reduce gastric acidity, impairing calcium and vitamin B12 absorption. Appetite suppression caused by medications like beta-blockers and certain antidepressants should also be evaluated using validated tools such as the Appetite, Hunger and Sensory Perception (AHSP) questionnaire, with scores below 40 indicating clinically significant anorexia warranting intervention [24].

Intervention strategies should prioritize simplifying medication regimens where feasible, collaborating with pharmacists to minimize pill burden and reduce adverse nutritional impacts. Nutritional supplementation may include high-protein oral nutritional supplements fortified with micronutrients, particularly vitamin D3 (cholecalciferol 1000–2000 IU/day) and B-complex vitamins, to compensate for malabsorption risks. Additionally, monitoring and managing gastrointestinal side effects such as nausea and constipation—common with multiple medications—are critical, utilizing prokinetic agents or fiber supplements as appropriate to maintain adequate nutrient intake.

Regular reassessment every 3 months is recommended to adjust nutritional plans in response to changes in medication regimens or clinical status. Multidisciplinary coordination between nephrologists, dietitians, pharmacists, and primary care providers is paramount to optimize both pharmacological and nutritional care, thereby improving patient outcomes and quality of life in CKD patients on complex multi-tablet therapies.

(5)
*Nutritional Intervention Using Oral Nutritional Supplements*


Oral Nutritional Supplements (ONS) are indicated in CKD patients who do not achieve adequate nutritional intake despite dietary counseling, particularly those meeting one or more of the following objective criteria: unintentional weight loss exceeding 5% within 1–3 months or >10% over 6 months (serum albumin persistently below 3.8 g/dL), dietary energy intake consistently less than 75% of recommended requirements, typically <25 kcal/kg/day, and protein intake below 0.8 g/kg/day in non-dialysis or below 1.2 g/kg/day in dialysis patients. Additionally, handgrip strength falling below 27 kg in men and 16 kg in women or evidence of muscle mass depletion by bioelectrical impedance analysis (e.g., phase angle < 5°) supports ONS initiation. Chronic inflammation indicated by elevated C-reactive protein (>5 mg/L) further substantiates the need for ONS due to increased catabolic demands [20,22,24,52] (Table 3).

In dialysis patients, who require higher protein intake (1.2–1.5 g/kg/day) to compensate for treatment-related losses, renal-specific ONS with controlled electrolytes—potassium < 300 mg and phosphorus < 250 mg per serving—such as Nepro HP or Novasource Renal are recommended. For non-dialysis CKD stages 3–5 patients, protein intake targets are 0.6–0.8 g/kg/day, with energy needs around 30–35 kcal/kg/day; low-protein, energy-dense supplements like Renilon 7.5 are suitable to meet these requirements while minimizing nitrogen load.

Expected benefits from ONS include prevention or reversal of PEW, demonstrated by weight stabilization or gain > 1–2 kg over 3 months, improvements in serum albumin by 0.2–0.4 g/dL, enhanced muscle strength, decreased hospitalization rates by approximately 20%, and lower mortality risk, especially in dialysis populations. Intradialytic ONS administration improves adherence and nutritional status without suppressing appetite. Thus, ONS are critical in mitigating malnutrition-related morbidity and mortality in CKD.

## 6. Collaboration and Patient Education in Nutritional Management of CKD

Addressing malnutrition stemming from nutritional insecurity in patients with CKD requires a comprehensive, multifaceted, and measurable approach. Central to this endeavor is ensuring patients consistently meet their minimum nutritional requirements, which demands both improved food access and systematic patient education [19].

Many CKD patients face financial and resource limitations that restrict their access to nutrient-dense, kidney-friendly foods—such as low-potassium fruits and vegetables, whole grains, and affordable, high-quality protein sources like eggs and legumes. Connecting patients with food assistance programs that provide these options is imperative. Moreover, education focused on budget-conscious meal planning—such as selecting frozen or canned low-sodium vegetables and incorporating plant-based proteins—empowers patients to make appropriate dietary choices within their economic means.

Beyond food access and education, effective nutritional management necessitates multidisciplinary collaboration involving physicians, registered dietitians, nurses, and social workers. Registered dietitians with expertise in renal nutrition serve as primary providers of personalized dietary counseling tailored to individual patient needs. Nephrologists, nurses, and social workers support this process by reinforcing dietary advice, managing comorbid conditions, and facilitating patient adherence. In resource-limited settings, trained peer educators and community health workers may supplement professional care to extend educational reach and engagement.

Social workers play a pivotal role in mitigating psychosocial barriers by linking patients with community resources, including food assistance programs, transportation, and financial support. Nurses contribute through ongoing monitoring and reinforcement of educational messages. This multidisciplinary team collaborates to assess nutritional status using validated tools and to formulate individualized care plans addressing not only dietary intake but also underlying factors such as inflammation, metabolic imbalances, and psychosocial challenges. Regular communication among team members facilitates timely modifications in nutritional interventions, thereby enhancing adherence and clinical outcomes (see Figure 1).

## 7. Who Should Provide Nutritional Education?

Nutritional education should primarily be delivered by registered dietitians with specialized expertise in renal nutrition, who can tailor dietary counseling to the unique needs of each patient. Supporting roles are fulfilled by nephrologists, nurses, and social workers who reinforce nutritional advice, manage comorbidities, and promote adherence to dietary plans. In settings with limited resources, trained peer educators and community health workers can expand the reach of educational efforts, ensuring wider patient engagement and support.

## 8. How Should Nutritional Education Be Delivered?

Effective education must be interactive, patient-centered, and adapted to the individual’s literacy level and psychosocial context. Combining verbal counseling with clear, accessible written materials enhances patient understanding. Visual aids such as portion control plates, food models, and digital applications are valuable tools for conveying complex dietary concepts. Both one-on-one counseling and group sessions provide distinct benefits: personalized problem-solving versus peer support and shared learning. Telehealth platforms have become increasingly important, especially during the COVID-19 pandemic, allowing flexible, continuous education and follow-up.

## 9. Frequency and Timing of Nutritional Education

Educational interventions should begin promptly at CKD diagnosis and intensify during critical clinical transitions, including progression to advanced CKD stages 4–5, dialysis initiation, and post-transplant periods. Follow-up sessions are recommended at regular intervals, typically every 3 to 6 months, with flexibility to increase frequency based on nutritional risk, laboratory data, or hospitalizations. This iterative process requires ongoing reassessment and adjustment of education strategies to accommodate changes in clinical status, cognitive function, and psychosocial factors, thereby optimizing adherence and improving patient outcomes.

Participation in professional workshops and conferences—such as the International Congress of Renal Nutrition and Metabolism, the ISRNM Total Nutrition Therapy program, and ESPEN courses—provides healthcare professionals with updated knowledge and best practices, further enhancing the quality of nutritional care.

By integrating evidence-based strategies with collaborative multidisciplinary teamwork and patient-centered education, healthcare providers can more effectively combat malnutrition caused by nutritional insecurity, ultimately improving both quality of life and prognosis for patients with CKD.

## 10. Emerging Technologies and Future Direction

The concept of precision nutrition is increasingly being introduced in the management of patients with CKD. Precision nutrition means an individualized approach in which dietary intervention are tailored according to patient’s genetic, biologic and environmental factors. It reflects genetic, metabolic, and or physiologic factors that affect how different patients respond to specific dietary interventions. In contrast to conventional nutritional management approach, which typically utilize a one-size-fits-all model, precision nutrition and personalized nutrition utilize an individualized approach. (e.g., genetic, microbiome, metabolic). Also, it considers factors such as lifestyle, environment, and patient’s preference. Precision nutrition is evolving rapidly with advances in data collection, deep data analysis, and modeling with application of artificial intelligence, while personalized nutrition is what we practice today leveraging available data and guidelines to meet each individual patient’s needs [54]. Nutri-genetics, nutrigenomics, microbiome profiling, metabolomics, and proteomics can be adjusted to guide dietary recommendations according to an individual’s unique characteristics [55].

There has been growing interest in how nutri-genetics, which studies the role of genetic polymorphism on the relationship between dietary patterns and phenotypes, affects CKD outcomes. For example, certain CKD patients may carry polymorphisms in genes related to vitamin D metabolism (such as the VDR gene), which can influence their vitamin D status and response to supplementation. By identifying these genetic variants, clinicians can tailor vitamin D dosing to optimize bone health and reduce cardiovascular risks more precisely than standard protocols. Furthermore, with microbiome profiling, CKD is associated with gut dysbiosis, which can worsen uremic toxin accumulation and inflammation [56,57]. Microbiome profiling identifies individual microbial compositions, enabling personalized dietary recommendations to promote beneficial bacteria [58,59]. For example, a CKD patient with low levels of short-chain fatty acid (SCFA)-producing bacteria might receive tailored prebiotic and fiber-rich diets to restore microbial balance and reduce toxin levels like indoxyl sulfate [53,60].

After all, advancing precision nutrition in kidney heath will require multidisciplinary collaboration and further study of the integration of advanced technologies such as AI that can harmonize complex data in order to tailor nutrition strategies that will optimize the health and well-being of the CKD population.

## 11. Conclusions

Malnutrition is highly prevalent among patients with CKD and has a detrimental impact on their prognosis; therefore, proactive efforts to improve nutritional status are essential. Nephrologists should pay attention to risk of malnutrition in their patients and to implement timely and proactive interventions aimed at nutritional improvement.

A multidisciplinary team for management is crucial in the management of malnutrition in CKD patients. The strategy to individualize dietary advice is most important. It would be desirable to involve heath care professionals working in nephrology and dialysis units, educated in screening for malnutrition, and a multidisciplinary team involving nephrologists, certified registered kidney dietitians, and social and community health workers.

The novel tools and emerging technologies to support nutritional assessment will ensure successful intervention and personalized medicine. Development of a diet quality tool would be valuable and improve understanding of the change in diet quality in response to dietary interventions. Large-scale, longitudinal randomized trials are also required to establish the clinical utility and cost-effectiveness of using proper nutritional assessment tools and dietary biomarkers, as well as the impact on patient quality of life, burden of care, and clinical decision making.

## Figures and Tables

**Figure 1 nutrients-17-03264-f001:**
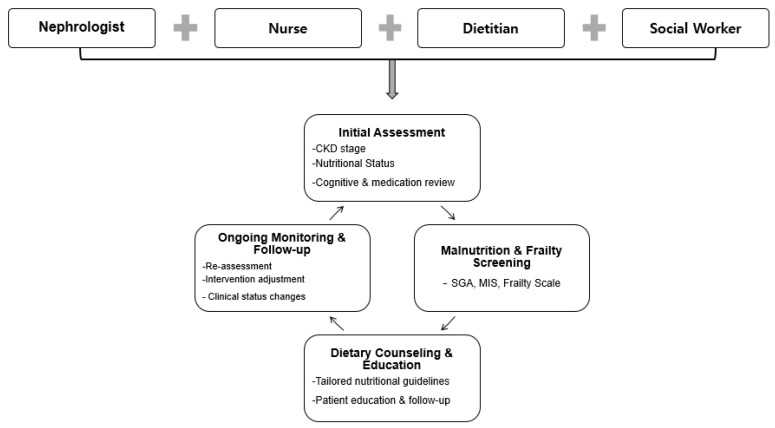
Multidisciplinary team for nutrition intervention. CKD, chronic kidney disease; MIS, malnutrition-inflammation score; SGA, subjective global assessment.

**Table 1 nutrients-17-03264-t001:** Summary of Nutritional Assessment Tools in CKD Patients.

Assessment Tool	Dialysis Patients	Non-Dialysis CKD Patients	Strengths	Limitation	Clinical Application
**BIA**	- Affected by fluid shifts during and after dialysis - No differentiation between **hydration and muscle mass** in overhydrated patients.	- **Early detection** of muscle wasting - Less studied in early CKD	- **Non-invasive** - **Fast and reproducible** - **Detects early malnutrition and muscle loss**	- Affected by hydration status - **Timing dependency in dialysis** - **Limited access and cost:**	- **Monitoring over time** - **Supports diagnosis of PEW and sarcopenia**
**SGA**	- **Confounded by fluid overload**, especially in PD patients. - Does not directly quantify muscle mass or intake.	- Early detection of PEW - Correlates with **eGFR decline** and inflammation markers - Less validated in early CKD stages.	- Correlates with morbidity and mortality; - Easy, low-cost	- Subjective; inter-rater variability - May miss early/mild malnutrition - Fluid overload can affect accuracy	- Suitable for routine use and monitoring nutritional trends
**MIS**	- Strong prognostic value - **Influenced by inflammation and comorbidities, not just nutrition.**	- Associated with **eGFR decline** and risk of PEW over time - **Less validated** in CKD stages 1–3.	- Includes lab markers and comorbidities - Better prediction of outcomes vs. SGA - Longitudinally responsive	- Time-consuming - Includes subjective items - May be influenced by inflammation more than malnutrition	- For tracking changes over time
**BMI**	- Measure post-dialysis dry weight - Watch for fluid overload	- Reliable but adjust cutoffs (e.g., <23 kg/m^2^ risk)	- Easy to measure and monitor - Can signal undernutrition (e.g., BMI < 18) - Associated with outcomes	- Cannot distinguish fat vs. muscle mass - Affected by fluid status - May mask sarcopenic obesity	- Screen for malnutrition/obesity - Monitor rends monthly
**HGS**	- Use non-fistula arm - <27 kg men, <16 kg women indicates sarcopenia	- Early sarcopenia detection - <20 kg men, <12 kg women risk	- Reflects muscle function and protein reserves - Correlates with outcomes and PEW - Inexpensive and quick	- Influenced by comorbidities (e.g., neuropathy) - Standardization of measurement is essential	- Monitor inflammation and nutritional status monthly
**Serum Albumin/Prealbumin**	- <3.8 g/dL risk - Consider inflammation	- <3.5 g/dL risk - Interpret with CRP	- Associated with mortality risk - Prealbumin responds faster to interventions	- Affected by inflammation, liver function, fluid status - Not specific for malnutrition	- Not recommended as standalone - Use as supplementary indicator
**nPCR**	- Target ≥ 1.0 g/kg/day - <0.5 indicates PEW	- Estimated via urine urea - 0.8–1.0 g/kg/day recommended	- Estimates dietary protein intake - Reflect nutrition-inflammation balance - Tracks change post-intervention	- Complex to calculate - Affected by residual renal function, catabolism, inflammation	- Guide protein intake - Assess intervention efficacy quarterly
**GLIM criteria**	- Use DXA for muscle mass	- Use weight loss and intake data - Inflammation important	- Combines phenotypic + etiologic criteria - Promotes global standardization	- Accuracy variable in CKD - Low concordance with SGA in some studies	- Screen at-risk patients - Every 3–6 months

BIA, bioelectrical impedance analysis; CKD, chronic kidney disease; HGS, handgrip strength; GLIM, global leadership initiative on malnutrition MIS, malnutrition inflammation score; PEW, protein-energy wasting; peritoneal dialysis; SGA, subjective global assessment.

**Table 2 nutrients-17-03264-t002:** Recommended Nutrient Intake by CKD Stage and Special Conditions.

Nutrients	CKD Stage 1–3	CKD Stage 4–5	Dialysis Patients	Frailty CKD Patients	Malignancy Patients with CKD
**Energy intake**	30–35 kcal/kg/day	30–35 kcal/kg/day	30–40 kcal/kg/day (adjusted for age and activity)	30–40 kcal/kg/day 40–45 kcal/kg/day in cachexia	30–40 kcal/kg/day 35–45 kcal/kg/day in hypermetabolic states or active malignancy
**Protein Intake**	0.8 g/kg/day Moderate restriction to slow CKD progression But avoid malnutrition	0.6–0.8 g/kg/day Monitored to avoid PEW	1.2–1.4 g/kg/day	At least 1.0–1.2 g/kg/day Up to 1.5 g/kg/day, to preserve muscle mass and function	1.2–1.5 g/kg/day Avoid excessive restriction
**Sodium Intake**	<2.3 g/day	<2.3 g/day	2.0 g/day or less	<2.3 g/day Monitor carefully due to frailty-related hyponatremia risk	<2.3 g/day Balanced with symptom control and patient tolerance
**Potassium Intake**	Usually unrestricted if serum K+ < 5.0 mEq/L	2.0–3.0 g/day if serum K+ > 5.0 mEq/L	Generally unrestricted due to dialytic removal Limit to 3.0–4.0 g/day based on serum K+ and dialysis adequacy	Individualized	Restrict as needed based on kidney function and chemotherapy-induced electrolyte disturbances
**Phosphorus Intake**	800–1000 mg/day Avoid excessive intake to delay mineral bone disorder	800 mg/day or less; Phosphorus binders often required as CKD progresses	800–1000 mg/day Phosphorus binders standard monitor PTH levels	800 mg/day or less Restriction with risk of protein malnutrition	800–1000 mg/day Minimize bone complications and tumor lysis syndrome

CKD, chronic kidney disease; PEW, protein-energy wasting.

**Table 3 nutrients-17-03264-t003:** ONS Supplementation and its outcomes and monitoring by CKD stage.

CKD Stage/Dialysis Status	ONS Indication	Recommended Protein Intake (Including ONS)	Key Nutrition Composition	Expected Outcomes	Monitoring Parameters	Assessment Interval
**CKD 3–5 (Non-Dialysis)**	- Inadequate dietary intake (<75% energy/protein targets) - Presence of PEW	- 0.55–0.60 g/kg/day protein - supplemented by ONS to meet targets without exceeding	- Renal-specific formulas - ~200–300 kcal - 10–15 g protein - Low Na, K, P content	- Improved serum albumin - Prevention of weight loss - Maintenance of muscle mass	- Serum albumin, prealbumin - Electrolytes (P, K) - Body weight - Muscle strength	Every 1–3 months
**HD patients**	- Persistent inadequate intake despite counseling - Presence of malnutrition or PEW	- 1.0–1.2 g/kg/day protein total - Including ONS	- Renal-specific ONS - ~300–400 kcal, - 5–25 g protein - Low P,K content	- Increased serum albumin - Improved BMI, lean body mass - Enhanced muscle strength - Reduced hospitalization rates	- Serum albumin - nPCR - Electrolytes (P,K) - Fluid status - Interdialytic weight gain - Muscle function	Every 1–3 months
**PD patients**	- Same as HD - Additional losses of protein via dialysis fluid increase needs	- 1.0–1.2+ g/kg/day including ONS	- Similar renal-specific ONS formula	- Better preservation of nutritional status - Muscle mass	As above plus peritoneal dialysis parameters (dialysate protein losses)	Every 1–3 months

CKD, chronic kidney disease; HD, hemodialysis; K, potassium; Na, sodium; ONS, oral nutritional supplement; PD, peritoneal dialysis; PEW, protein-energy wasting; P, phosphorus.

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
