# Peer review of "Nutritional Status Evaluation and Intervention in Chronic Kidney Disease Patients: Practical Approach"

_nutrients, 2025, doi:10.3390/nu17203264_

Round 1

Reviewer 1 Report

Comments and Suggestions for Authors

The manuscript presents a review comparing nutritional assessment tools for CKD
patients. However, the work suffers from substantial methodological, scientific, and
formatting issues that preclude consideration for publication in its current form.
1. The manuscript should clearly define how the specificity and sensitivity of each
nutritional assessment tool were determined. Additionally, the criteria for identifying
the advantages and limitations of each tool should be explained, with proper
scientific evaluation.
2. The authors should provide relevant information on how each nutritional assessment tool defines the stages of CKD.
3. In Table 2, there are typographical errors, and the use of symbols should be consistent. The meaning of the “~” symbol should be clarified, and the choice between “:” and “=” should be standardized.
4. The range “30–35 kg/kg/day” contains an error and should be corrected to “30–35
kcal/kg/day.” The unit “kcal” should be used consistently throughout the manuscript.
5. All cited references should appear before the period in the sentences. Additionally, Table 2 should include references for each category listed.
6. There is an absence of Figure 2; the current figure labeled as “Figure 2” appears to
be Figure 3.
7. The manuscript lacks scientific rigor, contains numerous typographical and
formatting errors, and fails to provide sufficient justification for its conclusions. 

Author Response

Dear Editor, Nutrients

Thank you very much for the evaluation of our manuscript. We are returning a revised manuscript which incorporates many of the suggestions made by four reviewers. A response to the referees΄ suggestions has been listed one by one, and an index of change has been included. We hope that the comments of the referees are adequately addressed in the revised manuscript.

Manuscript number: Nutrients-3809726

Manuscript title:

Nutritional Status Evaluation and Intervention in Chronic Kidney Disease Patients: Practical Approach

Index of changes 

Major changes:

  1. Addition of applicability of malnutrition tools in different clinical settings
  2. Alignment of content for consistency between abstract and introduction aim
  3. Addition of recent KIDGO content throughout the manuscript
  4. Addition of stratified interventions for special clinical conditions
  5. Provision of information of CKD stage by nutritional assessment tools
  6. Correct Table 1 ,2 with objective clarification
  7. Redrawing Figure 1.

Minor changes

  1. Correction of typos
  2. Standardization of units and symbols

Sincerely yours,

Eun Jeong Ko, MD, PhD

Division of Nephrology, Department of Internal Medicine, Bucheon St. Mary’s Hospital, College of Medicine, The Catholic University of Korea

327, Sosa-ro, Bucheon-si, 14647Gyeonggi-do, Korea

Fax: +82-32-340-7161, Phone: +82-32-340-7161

E-mail: neat0505@gmail.com

 Response to Editor’s Comments to Author:

The aims of the manuscript state to review the different tools to assess malnutrition. The manuscript itself outlines what some of these tools are but there is no critical appraisal of how these may be used in different settings etc. Additionally, the manuscript discusses dietary patterns and restrictive diets but this is not outlined in the introduction/ aims. Whilst a comprehensive review is useful some more critical appraisal of the tools and also different dietary recommendations is needed. There are recent KDIGO guidelines that mention dietary intake specifically and this should be incorporated.

→ We added the applicability of malnutrition tools in different clinical setting. Considerations for the application of each nutritional assessment tool across various clinical conditions – including CKD stage, dialysis status, frailty, and malignancy – are described, and a summary of their advantages and limitations is presented in revised version of Table 1 based on major clinical guidelines, including KIDIGO 2024 update.  Furthermore, revised Table 2 presents a detailed summary of the recommended nutrient intake levels—including energy, protein, sodium, potassium, and phosphorus—specified numerically according to CKD stage, dialysis status, and other special conditions such as frailty and malignancy. Additionally, we properly revised the content to ensure alignment between the abstract and the introduction properly.

Perhaps some more integration of malnutrition and frailty nutrition guidelines would be useful to integrate these recommendations into the review.

→ We reviewed literature related to malnutrition and frailty and provided specific recommendations for nutritional interventions, including those summarized in Table 2. Additionally, a section titled “Special Considerations in Assessing Nutritional Status of CKD Patients” was added to separately review the nutritional assessment and intervention strategies for frail CKD patients, based on the KDIGO 2024, KDOQI, ESPEN, and ISRNM guidelines.

Some spelling mistakes need to be rectified - tolls = tools

→ Typographical errors were corrected.

Editor 1 :
Comments to the Author:

Overall comments to the Author

The manuscript presents a review comparing nutritional assessment tools for CKD patients. However, the work suffers from substantial methodological, scientific, and formatting issues that preclude consideration for publication in its current form.

→ Acknowledging the valuable feedback, we have comprehensively revised the script to improve its methodological soundness and scientific clarity.

  1. The manuscript should clearly define how the specificity and sensitivity of each nutritional assessment tool were determined. Additionally, the criteria for identifying the advantages and limitations of each tool should be explained, with proper scientific evaluation.

→ We revised and reorganized the descriptions of each nutritional assessment tool, focusing on their clinical characteristics, strengths, and limitations, with particular attention to their applicability in specific clinical contexts. Furthermore, we comprehensively updated each item in Table 1 by incorporating key features of the tools based on multiple major guidelines and relevant studies. These revisions include specific quantitative values to facilitate practical application in clinical settings.

  1. The authors should provide relevant information on how each nutritional assessment tool defines the stages of CKD.

→ In agreement with the comments provided, we have revised the script to include detailed descriptions on how each nutritional assessment tool should be applied according to the specific stages of CKD. This content has also been summarized and organized in the revised Table 1 for clarity and ease of reference.

  1. In Table 2, there are typographical errors, and the use of symbols should be consistent. The meaning of the “~” symbol should be clarified, and the choice between “:” and “=” should be standardized.

→ All symbols mentioned in the comments have been standardized and revised accordingly.

  1. The range “30–35 kg/kg/day” contains an error and should be corrected to “30–35kcal/kg/day.” The unit “kcal” should be used consistently throughout the manuscript.

→ All units have been reviewed and standardized throughout the manuscript.

  1. All cited references should appear before the period in the sentences. Additionally, Table 2 should include references for each category listed.

→ We have carefully revised the reference formatting to ensure that all citations are consistently placed before the period at the end of the corresponding sentences. Furthermore, we have thoroughly revised Table 2, and the corresponding references have been appropriately cited in the manuscript wherever the table is referenced.

  1. There is an absence of Figure 2; the current figure labeled as “Figure 2” appears tobe Figure 3.

→ The figures have been comprehensively revised and consolidated into a single Figure 1. In addition, all previously incorrect figure references in the main text have been corrected accordingly.

Editor 2 :
Comments to the Author:

Overall comments to the Author

Ko and coworkers presented an interesting review: Nutritional status evaluation and intervention in chronic kidney disease patients: practical approach.

The authors addressed an interested topic, presented relevant previously published data, highlighted and discussed the literature, limitations and made the appropriate conclusions.

Closer look at the review led to the comments:

  1. The authors should define the chronic kidney disease (CKD) population more precisely. Throughout the text, the CKD population should be clearly defined and/or divided into non-dialysis dependent and dialysis-dependent CKD population.

→ We fully acknowledge the valuable feedback. Accordingly, the manuscript has been comprehensively revised to differentiate the application of assessment tools and interventions according to CKD stage and dialysis status, thereby providing a more practical and tailored approach.

  1. The practical approach in the title in is not sufficiently presented though the text.

→ Based on the valuable feedback received, the script has been revised throughout to place greater emphasis on practical aspects. Regarding nutritional assessment tools, their clinical application methods have been delineated according to CKD stage and dialysis status, grounded in the characteristics of each tool as described in major guidelines. In the intervention section, specific quantitative recommendations for nutrient supplementation according to CKD stage and dialysis status have been incorporated to facilitate practical implementation. Furthermore, a new section addressing special considerations—such as cognitive impairment, frailty, malignancy, amputation, and multi-tablet therapy—has been added to provide detailed guidance on nutritional approaches and interventions tailored to these specific conditions.

  1. The discussion about the role of oral nutritional supplements is missing.

→ In response to the comments provided, a separate section titled "Nutritional Intervention Using Oral Nutritional Supplements (ONS)" has been added. Additionally, Table 3 has been newly created to include a review of ONS supplementation, its outcomes, and monitoring according to CKD stage.

  1. The cognitive assessment of the patients is also missing.

→ In accordance with recommendation, we have incorporated a new section entitled “Special Considerations in Assessing Nutritional Status of CKD Patients,” within which a comprehensive review on “Cognitive Assessment in Nutritional Status Evaluation for CKD Patients” has been newly introduced.

  1. When and how to assess the BCM is the case of amputation?

→ We appreciate the insightful comment. Accordingly, detailed information regarding the methodology and interpretation considerations of bioelectrical impedance analysis (BIA) in cases of amputation has been added to the section titled “To Assess Body Composition: Bioelectrical Impedance (BIA).”

  1. Frailty and patients with malignancy should be discussed in more detail and separately.

→ We appreciate the important points raised. In the revised manuscript, we have added a new section titled "5. Special Considerations in Assessing Nutritional Status of CKD Patients" which includes additional reviews on "(2) Nutritional Assessment and Intervention in Frail CKD Patients" and "(3) Nutritional Assessment and Intervention in Malignancy CKD Patients."

  1. The problem of multi-tablet therapy should be highlighted and discussed

→  We concur with the valuable feedback and have accordingly added the section "5. Special Considerations in Assessing Nutritional Status of CKD Patients" wherein we present relevant content under "(4) Nutritional Assessment and Intervention in Multi-tablet Therapy."

  1. Who should be responsible for the patient education, how and how often the education necessary or should it be repeated?

→ In the revised section "6. Collaboration and Patient Education in Nutritional Management of CKD," we have addressed the important issue of who should provide nutritional education, how it should be delivered, and how frequently it should be conducted. Specifically, we expanded the discussion under the subheadings: "Who Should Provide Nutritional Education?", "How Should Nutritional Education Be Delivered?", and "Frequency and Timing of Nutritional Education." In addition, Figure 1 has been updated to present a comprehensive flowchart reflecting this educational framework.

  1. A separate figure of practical approach measures would be appropriate.

→ In accordance with your suggestion, we have revised Figure 1 to serve as a standalone illustration outlining the practical approach measures in nutritional management.

Reviewer 2 Report

Comments and Suggestions for Authors

Ko and coworkers presented an interesting review: Nutritional status evaluation and intervention in chronic kidney disease patients: practical approach.

The authors addressed an interested topic, presented relevant previously published data, highlighted and discussed the literature, limitations and made the appropriate conclusions.

Closer look at the review led to the comments:

  1. The authors should define the chronic kidney disease (CKD) population more precisely. Throughout the text, the CKD population should be clearly defined and/or divided into non-dialysis dependent and dialysis-dependent CKD population.
  2. The practical approach in the title in is not sufficiently presented though the text.
  3. The discussion about the role of oral nutritional supplements is missing.
  4. The cognitive assessment of the patients is also missing.
  5. When and how to assess the BCM is the case of amputation?
  6. Frailty and patients with malignancy should be discussed in more detail and separately.
  7. Th problem of multi-tablet therapy should be highlighted and discussed.
  8. Who should be responsible for the patient education, how and how often the education necessary or should it be repeated?
  9. A separate figure of practical approach measures would be appropriate.

The authors should accept and discuss these comments.

Author Response

(The authors gave the same response as above.)

Round 2

Reviewer 1 Report

Comments and Suggestions for Authors

All my comments were responded to.

Reviewer 2 Report

Comments and Suggestions for Authors

Ko and coworkers presented an interesting review: Nutritional status evaluation and intervention in chronic kidney disease patients: practical approach.

The authors addressed an interested topic, presented relevant previously published data, highlighted and discussed the literature, limitations and made the appropriate conclusions.

The authors correctly accepted the comments and improved the manuscript.